# Artificial Lighting Environment Evaluation of the Japan Museum of Art Based on the Emotional Response of Observers

**Zhisheng Wang [1,2], Yukari Nagai [1,*], Jiahui Liu [2], Nianyu Zou [2] and Jing Liang [2]**

[1] Knowledge Science, Japan Advanced Institute of Science and Technology, Nomi 923-1292, Japan; wangzs@dlpu.edu.cn

[2] Research Institute of Photonics, Dalian Polytechnic University, Dalian 116034, China; liujhhhhhh@163.com (J.L.); n_y_zou@dlpu.edu.cn (N.Z.); liangjing@dlpu.edu.cn (J.L.)

* Correspondence: ynagai@jaist.ac.jp; Tel.: +81-0761-51-1706

**Abstract:** This paper mainly studies the effect of artificial lighting environmental factors on the psychological emotions of observers in the large and practical space of the museums. The purpose is to reveal the relationship between the observers' response and the artificial lighting condition in the actual art museum space. Field research regarding three art museums in Japan was carried out and the optical environment parameters applied in those museums were quantified. The innovation method is to define the artificial lighting environment space in the way of classified lighting design. Thirty one observers were invited to evaluate the three art museum's lighting environment. In addition, this paper analyzes and discusses the influence of the actual spatial lighting parameters of museum buildings on observers' psychological emotions (comfort, clarity, preference and warmth), under three modes of illuminance and correlated colour temperature (CCT) combination. Using one-way analysis of variance and correlation analysis, through analysis get the correlation of the four evaluation and three lighting environments indexes are less than 0.05, the observer in an environment with high illuminance and a high CCT had higher psychological evaluation of the art museum.

**Keywords:** photometric and colorimetric; museum lighting; psychological perception; correlated colour temperature (CCT); illuminance

## 1. Introduction

Museums aim to exhibit, protect and educate. The first two are usually competing with each other [1]. For example, in a bid to achieve higher display quality, higher visual sensitivity is often required, which may damage exhibits, especially with the two spectral components of ultraviolet and infrared in light [2]. At the time when building the major picture galleries and museum, artificial lighting was still in its infancy [3].

Museum design, as a topic of great interest for interior designers, architects, lighting designers and museum personnel, is increasingly attracting the public. Despite the activity in museum design, the large sums of money spent to acquire artifacts and design buildings to house, display, protect collections and the importance of good lighting to successful design, rigorous analyses of the museum lighting environment are extremely limited [4]. The lighting design of exhibition space has a great impact on visual and colour perception and different lighting arrangements can create quite different visual impressions of artworks and, if not exquisitely designed, might compromise the enjoyment of viewers [5].

The light quality is the foundation of the light source applied in each space environment. Only when the quality of light is guaranteed, can we get better feedback from the observers. Huang

and colleagues published a series of papers on illumination whiteness and color preference, which described the relationship between perceived whiteness and color preference of lighting [6,7]. Their works proved that people did prefer the color rendition of white lighting at multiple correlated colour temperatures(CCTs) ranging from 2500 K to 5500 K, while they seemed to dislike colors that were illuminated by too cold CCT higher than 5500 K. The same group did two experiments, one involves evaluating color preferences for lighting with empty light box [8], to study color preferences for several LED white lights with different CCTs. The other studies the dominance of colour preference when the CCTs are different [9]. Lighting color preference is usually influenced by three environmental factors: light, object and observer. In that research, a series of psychophysical experiments were carried out to investigate and compare the effects of certain factors on color preference, including spectral power distribution of light, lighting application, individual color preference of observers, regional cultural differences, as well as gender differences.

Luo et al. studied LED lighting conditions suitable for viewing art in a museum environment, to test Kruithof's rule that defines pleasant lighting in terms of CCT and illuminance. Experimental results reveal that illuminance has a greater effect on the works than CCT. That is, an increase in illuminance will improve the scores of most scales. They found that "visibility" and "warmth" were the general feelings when viewing the painting, and developed two design-aware regional maps emotion-response models for museum space, applied to lighting in small and large spaces. Experiment 1 was to investigate the impact of illuminance conditions on visual perceptions in a light cabinet. Experiment 2 was the same as Experiment 1, except that it was performed in a museum. The results display that the visibility model is only related to illuminance and the warmth model is only related to CCT [10].

Scuello et al. studied the effects of light source CCT in museums. Experiments were in different rooms, each of which was independently illuminated by a colour-corrected tungsten light source and equipped with a neutral density filter to control the lighting. Among the 11 colours with temperatures ranging from 2500 K to 7000 K, the illuminance of a painting is 200 lx–250 lx. It shows that the observer is satisfied with the CCT at 3600 K [11].

Davis and Ginthner both disagreed with the Kruithof's rule [12]. In a colour-balanced environment with a colour rendering index (CRI) of approximately 90, it indicates that the subjective preference score is only affected by the illuminance level of 270 lx to 1345 lx, while not affected within the CCT range of 2750 K to 5000 K.

Yoshizawa et al. and Luo et al. made visual evaluation experiments on paintings under LED lighting [13,14]. It is concluded that visual perception is subject to visibility and texture (warm) when viewing museum paintings. In the model room and exhibition room of the Morohashi museum of modern art in Japan, two independent experiments were done. Yoshizawa et al. researched the changes of CCT (from 2700 K to 5000 K), CRI (from 55 to 100) and illuminance (up to 400 lux) respectively on LED illuminance.

TQ Khanh et al. studied the user preference model for interior lighting. By assessing their visual impressions about scene brightness (SB), visual clarity (VC), Colour preference (CP) and Scene preference (SP) try to find "good" levels of the visual attributes. The results show that criterion illuminance levels for "good" levels of the visual attributes were determined depending on CCT [15,16]. The equation for CP (1) and SP (2) are as below.

This is an example of an equation:

$$CP = 14.089 ln(E_v, eq) - 25.397 \tag{1}$$

CP in Equation (1) is the Colour preference. *Ev, eq* is the equivalent illuminance, *ln* is a logarithmic function. The mean visually scaled CP is depicted as a function of the model equation [15,16].

This is an example of an equation:

$$SP = 17.127 ln(E_v, eq) - 41.844 - 0.1325 \Delta C*^2 + 0.2797 \Delta C*$$
$$+ [-622.378 ((S/V)^{0.24})^2 + 980.843 (S/V)^{0.24} - 382.535] \tag{2}$$

SP in Equation (2) is the Scene preference, $\Delta C^*$ is saturation. *Ev, eq* is the equivalent illuminance. *ln* is a logarithmic function. The symbol *S* in Equation (2) represents the signal of the short-wavelength sensitive human photoreceptors obtained by weighting the relative spectral power distribution of the light source with the spectral sensitivity of the S-cones and integrating over the visible wavelength range. The quantity *V* is obtained by weighting the relative spectral power distribution of the light source with the *V(k)* function and integrating over the visible wavelength range [15,16]. According to the equation, the values of CP and SP are calculated to analyze and discuss the colour preference and scene preference.

## 2. Experiment

This experiment differs from the control variable method adopted by most scientific research articles. It does not strictly control one variable to ensure the controllability of other factors and to strictly explore the relationship between various parts of the variable by changing only one variable. The paper is mainly to study the four different dimensions in a large space environment for observers to experience the psychological feelings of museum lighting, it is not a separate investigation of CCT and illuminance changes, but specific indicators in the actual museum environment are used for investigation and research. Correlation analysis and one-way analysis of variance were used to analyze the differences in four dimensions under three different lighting environments. In terms of the visual assessment temperature, based on the existing research results, we begin from four dimensions: comfort, clarity, preference and warmth.

The main survey of the comfort level is the coordination of the overall space light and shadow of the museum exhibition space, as well as the degree of psychological colour. Clarity checks whether the details and texture of the exhibits can be clearly displayed, which is satisfactory. Preference is that the overall artistic effect of lighting is outstanding, there are wonderful lighting performance effects and whether the lighting environment in which you are feeling the like. The warmth refers to the contrast of warmth and coldness on the senses, whiter means colder and yellower means warmer.

### 2.1. Illuminance Test Method

In the illuminance test we use the center point method. In the illuminance measurement area, it is generally divided into rectangular grids while the grid should be square. The illuminance shall be measured at the center of the rectangular grid, as in Figure 1.

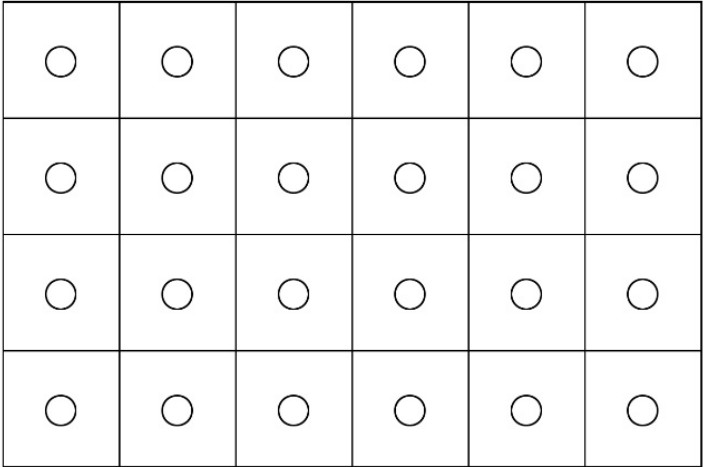

**Figure 1.** Lighting center arrangement.

This is an example of an equation:

$$E_{av} = \frac{1}{M \times N} \sum E_i \tag{3}$$

*Eav* in Equation (3) is the average illuminance in lux(lx). *Ei* is the illuminance of the *i* point in lux(lx). *M* refers to the longitudinal measuring point and *N* is the lateral measurement point [17]. The equipment used in the experiment to measure illuminance and CCTs is Konica Minolta CL-200A, and the CRI is measured by an Asensetek ALP-01 Pro.

### 2.2. Museum Lighting Environment

This experiment adopts lighting environment parameters of three museums to study the factors affecting psychological perception via different lighting methods (Only the lighting environment and typical paintings and background were evaluated). Dang R et al. studied the effects of illumination on inorganic pigments used in traditional paintings, they think that high illuminance could be a problem for some museum objects including textiles, tempera paints, etc. [18].

It selected 31 observers, aged 20–24 years. The first museum is the national museum of western art Figure 2a. The measuring is the painting exhibition room of the 19th–20th century. The illuminance is direct illuminance, CCT = 2079 K, x = 0.5239, y = 0.4208, colour rendering index Ra = 96, R9 = 92, average illuminance Eav = 84 lx. The second museum is the Aichi art museum Figure 2b. The measuring is the permanent exhibition room. The illuminance method is indirect illuminance, CCT = 2960 K, x = 0.4409, y = 0.4073, colour rendering index Ra = 90, R9 = 46, average illuminance Eav = 115 lx. The third museum is the Yamazaki Mazak art museum Figure 2c. The measuring is the painting exhibition room on the fifth floor. The lighting method is mixed lighting, CCT = 3568 K, x = 0.4130, y = 0.4196, colour rendering index Ra = 93, R9 = 88, average illuminance Eav = 538 lx. The parameters of CCTs and illuminance in the three lighting environment experimental spaces is within Kruithof's rule [19].

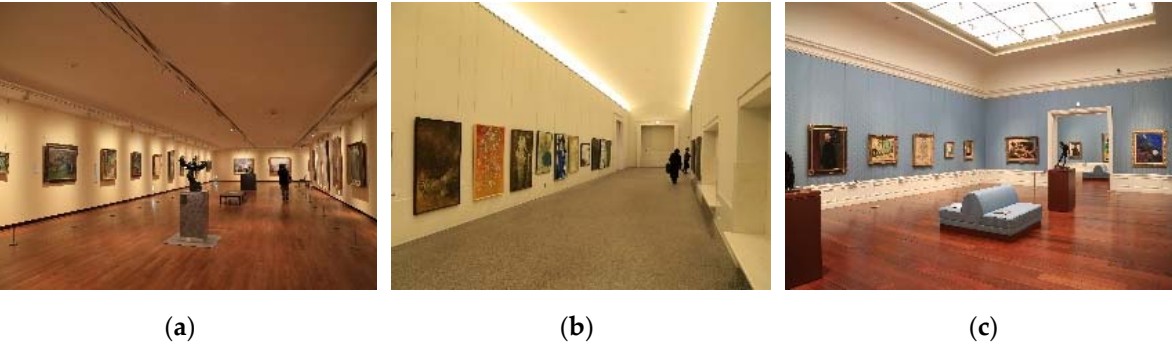

(**a**)　　　　　　　　　　(**b**)　　　　　　　　　　(**c**)

**Figure 2.** Three museum lighting environments. (**a**) The national museum of western art. (**b**) The Aichi Prefectural museum of art. (**c**) The Yamazaki Mazak museum of art.

CCTs analysis of the three lighting environments is shown in Figure 3. The first and second lighting environments CCT light color slant yellow, giving us a warm feeling. The CCT of the third lighting environment is white, which is a warm white light.

CRI analysis of the three lighting environments is shown in Figure 4. The R9 of the three art museums is 46, 92 and 88. CCTs, x and y are used to describe the CIE1931 chromaticity. The three spaces use different light sources and spectra, this method is used to define and quantify the parameters of the art museum lighting environment. And the CCTs in art museum lighting is different, which brings different visual experience and feeling to the observer. The lower the CCT, the warmer the hue of the art museum space; the higher the CCT, the cooler the hue of the art museum space and different CCTs give different subjective feelings. In order to accurately represent the colour of the lighting environment. The study uses the x and y values to give the position of the Correlated colour temperatures (CCTs) in CIE1931 accurately and provide more detailed optical data.

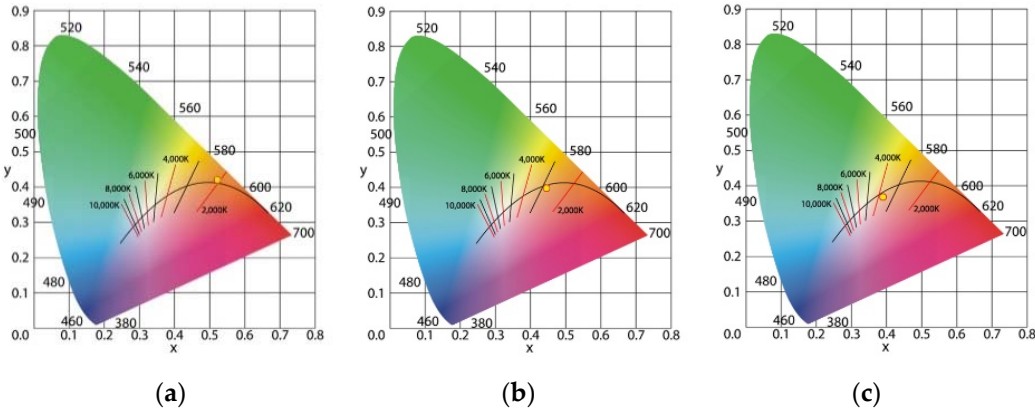

**Figure 3.** Colour coordinate diagram of three lighting environments. (**a**) Direct lighting CCT = 2079 K. The national museum of western art. (**b**) Indirect lighting CCT = 2960 K. The Aichi Prefectural museum of art. (**c**) Mixed lighting CCT = 3568 K. The Yamazaki Mazak museum of art.

The R9 is a very important evaluation index for museum lighting. R9 is saturated red, it is an indicator of the ability of the light source to restore the redness of the object. The larger the value of R9, the higher the ability of the light source to reduce the redness of the object. The LED light source used in most museums is blue light to excite yellow phosphors to emit light. The value of the red light spectrum in the spectrum of this light source is relatively low, but the red spectrum is important in the lighting of art museums, the R9 value was used as an important indicator of evaluating the quality of the lighting environment.

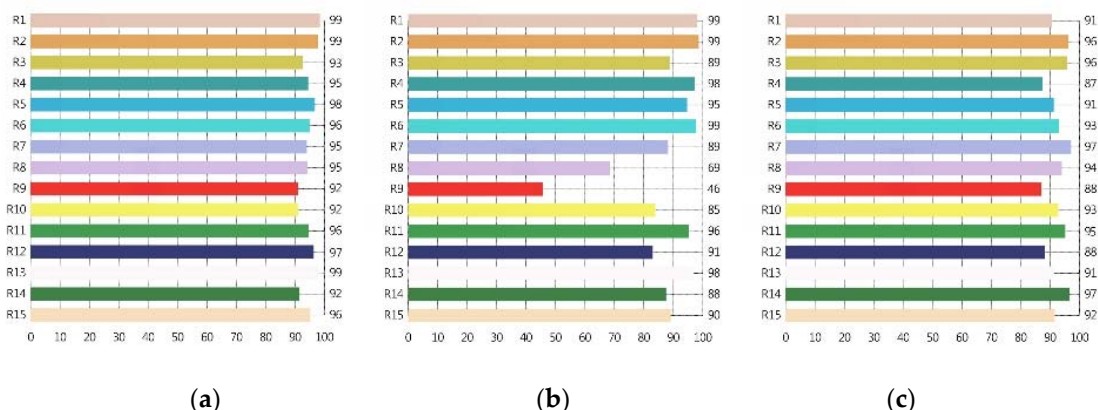

**Figure 4.** Colour rendering index(CRI) of three lighting environments. (**a**) Direct lighting Ra = 96, R9 = 92. The national museum of western art. (**b**) Indirect lighting Ra = 90, R9 = 46. The Aichi Prefectural museum of art. (**c**) Mixed lighting Ra = 93, R9 = 88. The Yamazaki Mazak museum of art.

People have maintained close ties with architecture. The size of the space inside a building also determines whether the lighting environment is comfortable for us. Good lighting design is inseparable from understood its architectural and space environment. The research classified the types of space, lighting and exhibits of three art museums. This type of research approach can be better defining the lighting environments of the space and exhibits.

The illuminance distribution in the national museum of western art lighting environment is shown in Figure 5a, the arrangement and elevation of the direct lighting display source is shown in Figure 5b. Direct lighting is adopted in the lighting environment. The lights shine directly into the middle of the paintings in the museum, highlighting the position of the paintings. The national museum of western art is installed with track on one side of the roof area by use of spotlights for direct lighting is shown in Figure 5.

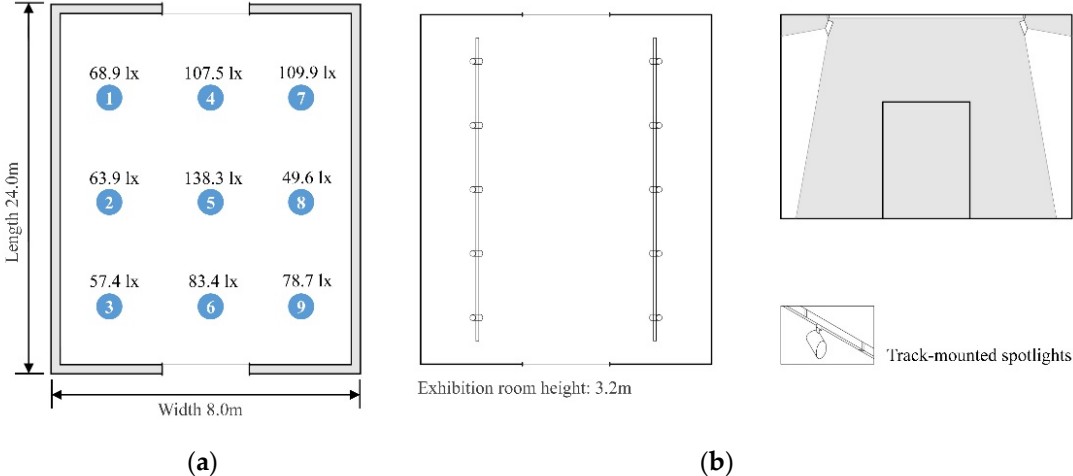

**Figure 5.** The national museum of western art building interior. (**a**) Illuminance plane spot test. (**b**) The arrangement and elevation of the direct lighting display source.

Figure 6 shows the paintings used in the national museum of western art lighting experiment, shows the 13 illumination distribution test points include typical oil painting and background wall. It was drawn by Claude Monet from French, an impressionist painter. The oil painting title is morning on the seine. The oil painting size is 82.0 × 93.0 cm, painting date in 1898. The average illumination of the oil painting and background wall is 225.4 lx, the minimum illumination is 180.2 lx, and the uniform of illumination is 0.80.

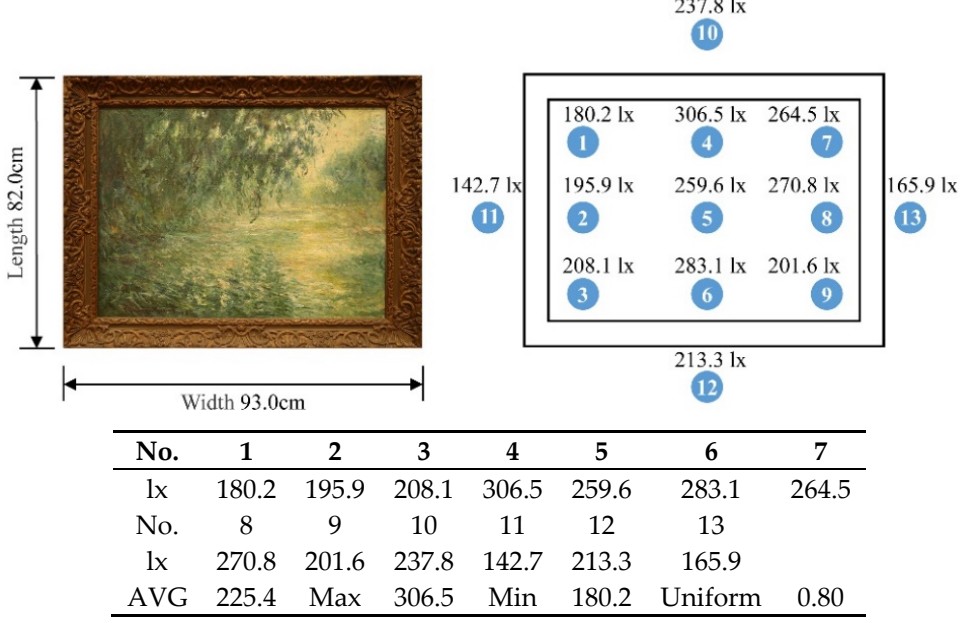

| No. | 1 | 2 | 3 | 4 | 5 | 6 | 7 |
|-----|-----|-----|-----|-----|-----|-----|-----|
| lx | 180.2 | 195.9 | 208.1 | 306.5 | 259.6 | 283.1 | 264.5 |
| No. | 8 | 9 | 10 | 11 | 12 | 13 | |
| lx | 270.8 | 201.6 | 237.8 | 142.7 | 213.3 | 165.9 | |
| AVG | 225.4 | Max | 306.5 | Min | 180.2 | Uniform | 0.80 |

**Figure 6.** Illumination distribution test of oil painting and background wall in indirect lighting environment.

The illuminance distribution in the Aichi Prefectural museum of art lighting environment is shown in Figure 7a, the arrangement and elevation of the indirect lighting display source is shown in Figure 7b. Indirect lighting is adopted in the lighting environment. The light source shines on the ceiling, illuminating the entire lighting environment with reflected light. The Aichi Prefectural museum of art has LED lamps arranged roof with indirect lighting is shown in Figure 7.

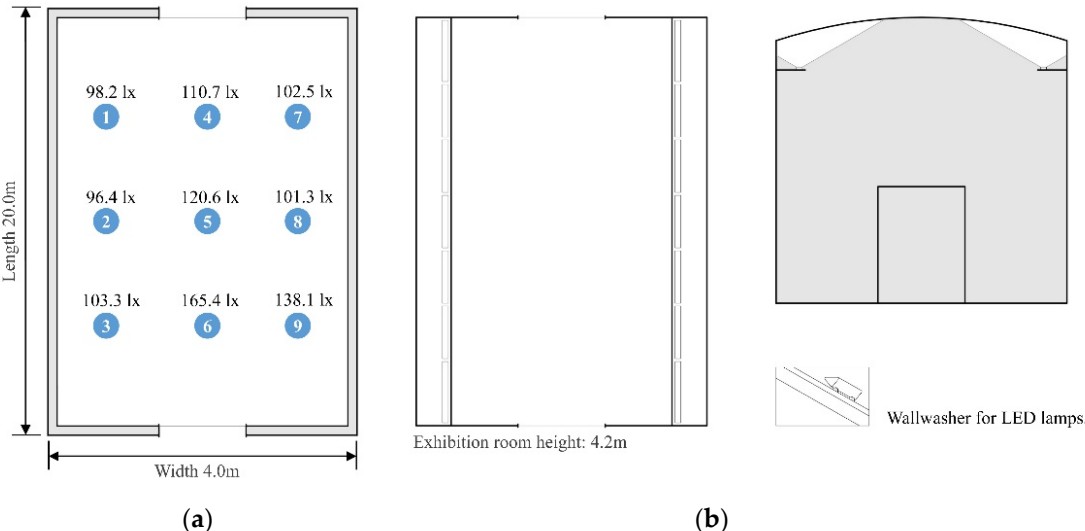

**Figure 7.** The Aichi Prefectural museum of art building interior. (**a**) Illuminance plane spot test. (**b**) The arrangement and elevation of the indirect lighting display source.

Figure 8 shows the paintings used in the Aichi Prefectural museum of art lighting experiment, shows the 13 illumination distribution test points include typical oil painting and background wall. It was drawn by Rinen Hoshi from Japan, a contemporary painter. The oil painting title is spring colorful flowers. The oil painting size is 180.6 × 86.4 cm, painting date in 2016. The average illumination of the oil painting and background wall is 117.6 lx, the minimum illumination is 148.5 lx, and the uniform of illumination is 0.84.

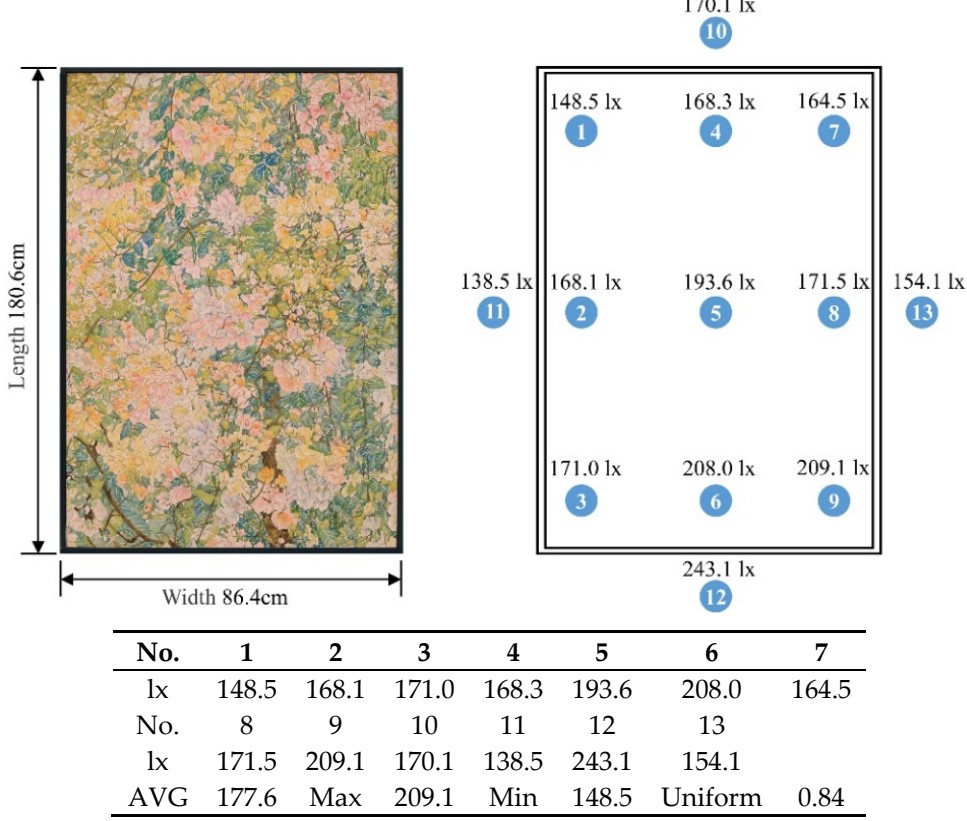

| No. | 1 | 2 | 3 | 4 | 5 | 6 | 7 |
|-----|-----|-----|-----|-----|-----|-----|-----|
| lx | 148.5 | 168.1 | 171.0 | 168.3 | 193.6 | 208.0 | 164.5 |
| No. | 8 | 9 | 10 | 11 | 12 | 13 | |
| lx | 171.5 | 209.1 | 170.1 | 138.5 | 243.1 | 154.1 | |
| AVG | 177.6 | Max | 209.1 | Min | 148.5 | Uniform | 0.84 |

**Figure 8.** Illumination distribution test of oil painting and background wall in indirect lighting environment.

The illuminance distribution in the Yamazaki Mazak museum of art lighting environment is shown in Figure 9a, the arrangement and elevation of the mixed lighting display source is shown in Figure 9b. Mixed lighting is used in the lighting environment. Both the light source directly illuminating the painting and the light source illuminating the ground are combined. The Yamazaki Mazak museum of art is equipped with an average arrangement for LED recessed downlights. A series of singlets allow additional accent lighting using spotlights is shown in Figure 9.

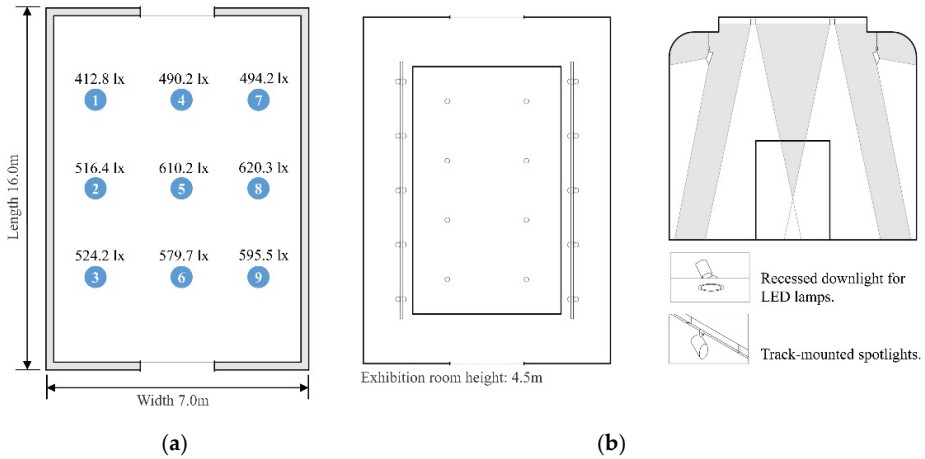

(**a**)                                                                (**b**)

**Figure 9.** The Yamazaki Mazak museum of art building interior. (**a**) Illuminance plane spot test. (**b**) The arrangement and elevation of the mixed lighting display source.

Figure 10 shows the paintings used in the Yamazaki Mazak museum of art lighting experiment, shows the 13 illumination distribution test points include typical oil painting and background wall. It was drawn by Jean-Marc Nattier from Paris, a rococo painter. The oil painting title is sailboat in the morning. The oil painting size is 101.8 × 82.8 cm, painting date in 1739. The average illumination of the oil painting and background wall is 381.9 lx, the minimum illumination is 331.4 lx, and the uniform of illumination is 0.87.

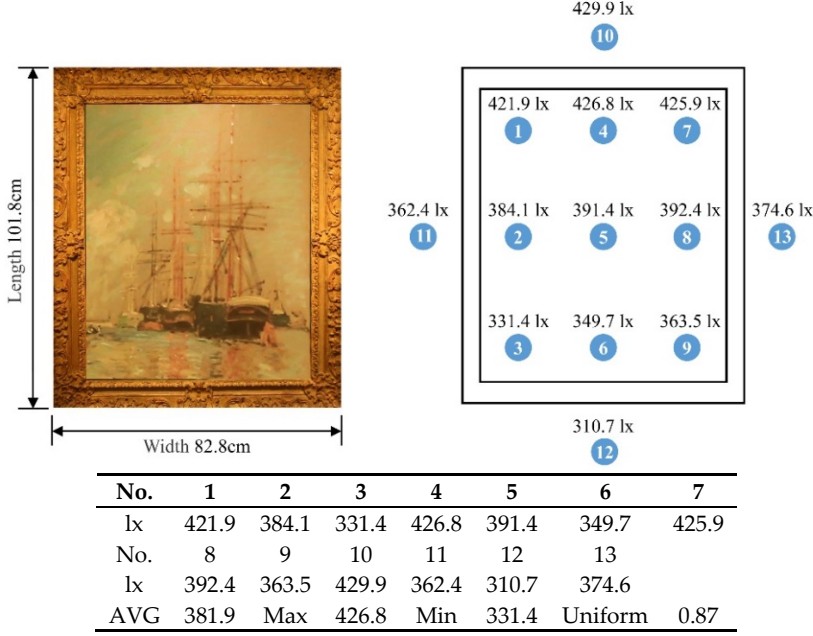

| No. | 1 | 2 | 3 | 4 | 5 | 6 | 7 |
|-----|------|------|------|------|------|------|------|
| lx | 421.9 | 384.1 | 331.4 | 426.8 | 391.4 | 349.7 | 425.9 |
| No. | 8 | 9 | 10 | 11 | 12 | 13 | |
| lx | 392.4 | 363.5 | 429.9 | 362.4 | 310.7 | 374.6 | |
| AVG | 381.9 | Max | 426.8 | Min | 331.4 | Uniform | 0.87 |

**Figure 10.** Illumination distribution test of oil painting and background wall in mixed lighting environment.

*2.3. Psychophysical Experiment*

Luo et al. did two experiments. Experiment 1 was to study the influence of lighting conditions on visual perception in the light box and proposed the semantic differential scale of 11 adjective pairs, which are designed to be related to perceived appearance and atmospheric perception. Experiment 2 was made in an actual museum and 10 adjectives were proposed [10]. Based on the 10 adjectives, this paper extracts 4 semantic differences about the environmental atmosphere, which is comfort, clarity, preference and warmth.

This experiment applies the method of categorical judgment in the psychophysical experiment method. The observer performs subjective scoring of comfort, clarity, preference and warmth in three different lighting environments.

A minimum of 1 point means completely not comfort/clarity/preference/warmth, a score of 2 means not comfort/clarity/preference/warmth, a score of 3 means less comfort/clarity/preference/warmth, a score of 4 means relatively comfort/clarity/preference/warmth, a score of 5 for comfort/clarity/preference/warmth, and up to a score of 6 for very comfort/clarity/preference/warmth. Table 1 gives an example for scoring comfort level, the description of the scores and the meaning of each category.

**Table 1.** An example of scoring (Comfort levels score).

| Choice by Observer SCORE | Comfort 1 | Comfort 2 | Comfort 3 |
|---|---|---|---|
| | 1 | 2 | 3 |
| Meaning | Extremely not comfort | Not comfort | Less comfort |
| Choice by observer Score | Comfort 4<br>4 | Comfort 5<br>5 | Comfort 6<br>6 |
| Meaning | Relatively comfort | Comfort | Very comfort |

Figure 11 shows the exhibition space of the art museum is classified in terms of architectural functions, it is mainly divided into viewing space, traffic space and rest space. Observers have different visual focuses in different spaces. In the rest space and the traffic space, the observer's visual focus is mainly the overall space, use the illumination of the ground in the lighting environment was used for evaluation. The main task in the viewing space is to appreciate artistic works, observer's visual focus is on the background walls and oil paintings, use the typical oil paintings and background wall illumination were used for evaluation.

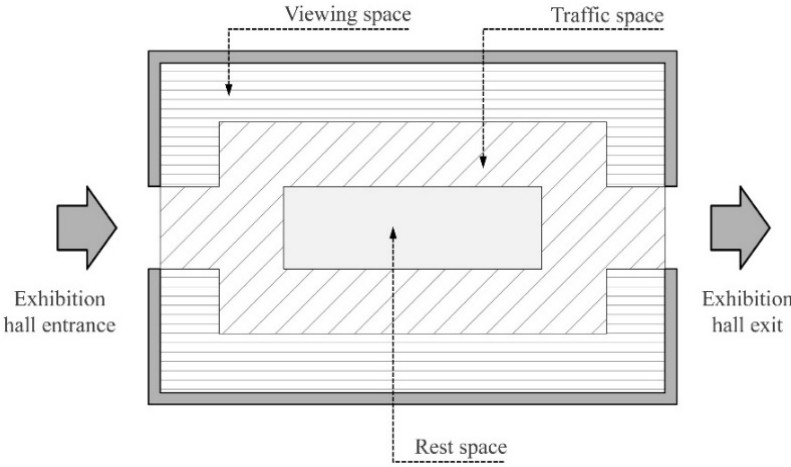

**Figure 11.** Typical exhibition space allocation in the museum of art.

The experimental process is as below:

1.　Introduction. Introduce the content and process of the art museum lighting experiment to the observer.
2.　Visual testing. Test the observer for visual defects or color blindness.
3.　Environmental adaptation. The observer first adapts to the museum lighting environment under a typical space for about 2 min. The experiment should allow the observers to whole experience the art museum lighting environment of the exhibition space and evaluate and feel the lighting environment of the viewing space, traffic space and rest space through observers' vision.
4.　Main experiment. Through the moving, viewing and staying of the observer in the exhibition space visual task is completed, the process takes about 5 min. Next, observers viewing the typical painting lighting environment in the viewing space, observe the oil paintings and feel the lighting environment in the viewing space, the process takes about 3 min.

Figure 12 shows the observation distance of the observers when viewing the oil painting. A limit is set before each painting to control the viewing distance to ensure that the observer's sightline is at the best viewing angle and distance. The typical size of the oil painting used in this experiment is suitable, so the observers are looking to have the optimal visual distance.

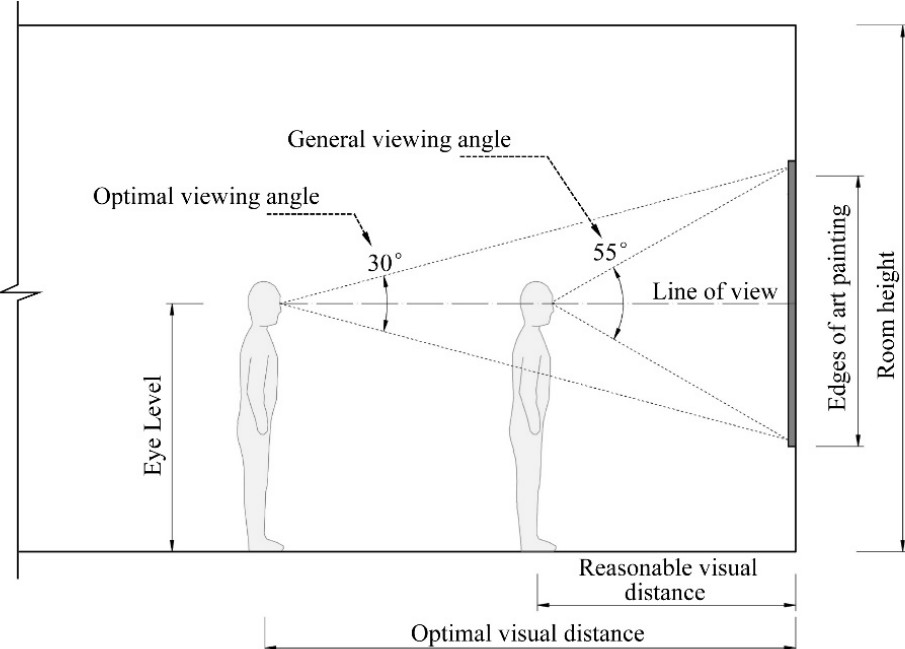

**Figure 12.** The relationship between painting and observers' visual distance.

5.　Subjective evaluation. The observer makes a subjective evaluation of the current lighting environment in the art museum, the observer fills in the corresponding score according to the questionnaire. The experimental process obtained subjective response data through four evaluation dimensions (comfort, clarity, preference and warmth) from observers. And analyzes the influence of the actual spatial lighting parameters of art museum buildings on observers' psychological emotions.
6.　Museum lighting environmental changes.
7.　Repeat steps 3–5 until the experimental data are collected.

*2.4. Observer Difference Analysis*

The same 31 observers selected went to three museums for field research and data evaluation. Each was evaluated for comfort, clarity, preference and warmth in three lighting environments, and 93 groups of subjective evaluation data were obtained. In this experiment, the Coefficient of variation (CV) is adopted to compare the data stability among observers [20].

This is an example of an equation:

$$CV = 100\sqrt{\frac{1}{N}\sum_{i=1}^{N}\frac{\left(X_i - \overline{X}\right)^2}{\overline{X}}} \tag{4}$$

In Equation (4), the score for the *i*-th observer is $X_i$, the mean for all observers is $\overline{X}$, $N$ is the number of observers, where $N = 31$.

Table 2 indicates the CV for the evaluation scores in different lighting environments. Environment 1 is the national museum of western art. Environment 2 is the Aichi Prefectural museum of art. Environment 3 is the Yamazaki Mazak museum of art. It's verified that the data in Table 2 are within the normal limits of the psychophysical experiment CV, the data obtained from the psychophysical experiments in this study are all reliable.

**Table 2.** CV for evaluating scores in different lighting environments.

| CV | Comfort | Clarity | Preference | Warmth |
|---|---|---|---|---|
| Environment 1 | 19.86 | 17.87 | 16.58 | 17.23 |
| Environment 2 | 18.92 | 14.05 | 14.96 | 13.81 |
| Environment 3 | 13.39 | 13.54 | 12.41 | 20.16 |

## 3. Experimental Results and Discussion

Through analyzing the spectral power distribution of the actual lighting environment of the three art museums and considering illuminance and CCT, the ΔC* is obtained by calculation (measured in terms of the quantity ΔC*, an object saturation measure computed in CIELAB colour space that corresponds to the mean value of the individual ΔC* values of the 15 CQS test colour samples VS1–VS15) [21]. It is observed in Table 3, three nominal illuminance levels (84 lx, 115 lx, 538 lx) and three nominal CCT levels (2188 K, 2903 K, 3671 K) were used. All of them were at the same high CRI (90 ≤ Ra ≤ 96) level corresponding to a low oversaturation level (−0.08 ≤ ΔC* ≤ 1.08).

**Table 3.** Three museums of 3 spectral properties. (No.1 is the national museum of western art. No.2 is the Aichi Prefectural museum of art. No.3 is the Yamazaki Mazak museum of art).

| No. | CCT (K) | ΔC* | Illuminance (lx) | CP | SP | Ra |
|---|---|---|---|---|---|---|
| 1 | 2188 | −0.08 | 84 | 7.82 | 8.96 | 96 |
| 2 | 2903 | 1.08 | 115 | 29.89 | 30.11 | 90 |
| 3 | 3671 | 0.41 | 538 | 61.17 | 63.98 | 93 |

By assessing their visual impressions about CP and SP, try to find "good" levels of the visual attributes. Under the actual horizontal illuminance of the light source, their visual impression on the scene brightness was evaluated, including CP and SP. The data in Table 3 indicates SP and CP. Distinctly, samples 3 > 2 > 1 demonstrate that the values of CP and SP of environment 3 are good.

From the perspective of correlation, the Pearson correlation coefficient between subjective preference score and the calculated results of the index (CP, SP) was calculated. The *p*-value was adopted to indicate whether the correlation was significant.

As shown in Table 4, the *p*-value of CP and SP was less than 0.05, indicating a high significance level. SP < 0.01 shows that it was more significant, and SP accounted for more of the preference score. The Pearson correlation coefficients(r) of CP and SP were both higher than 0.99, indicating that their preference scores were significantly correlated with CP and SP.

**Table 4.** Pearson correlation coefficient between preference rating and index (CP, SP).

| Name | r | *p*-Value |
|------|------|-----------|
| CP | 0.9996 | 0.0172 |
| SP | 1.0000 | 0.0040 |

One-way analysis of variance was carried out on the observer's evaluation in the three lighting environments. The results are demonstrated in Table 5.

The results in Table 5 indicate that the significance is less than 0.01 under the four indicators of comfort, clarity, preference and warmth, which demonstrates a significant difference in the observer's evaluation of comfort, clarity, preference and warmth in three lighting environments. Analysis of Variance (ANOVA) used to test the significance of difference between two or more samples.

**Table 5.** Results of one-way ANOVA of different psychological perceptions in three lighting environments.

| Evaluation Items | F Value | Significant |
|------------------|---------|-------------|
| Comfort | 14.569 | 0.000 |
| Clarity | 52.928 | 0.000 |
| Preference | 16.506 | 0.000 |
| Warmth | 14.093 | 0.000 |

In three different lighting environments, the four psychological factors (comfort, clarity, preference, warmth) are processed in relation to the data. Figure 13 shows the mean of the psychological perception of the 31 observers in these three lighting environments.

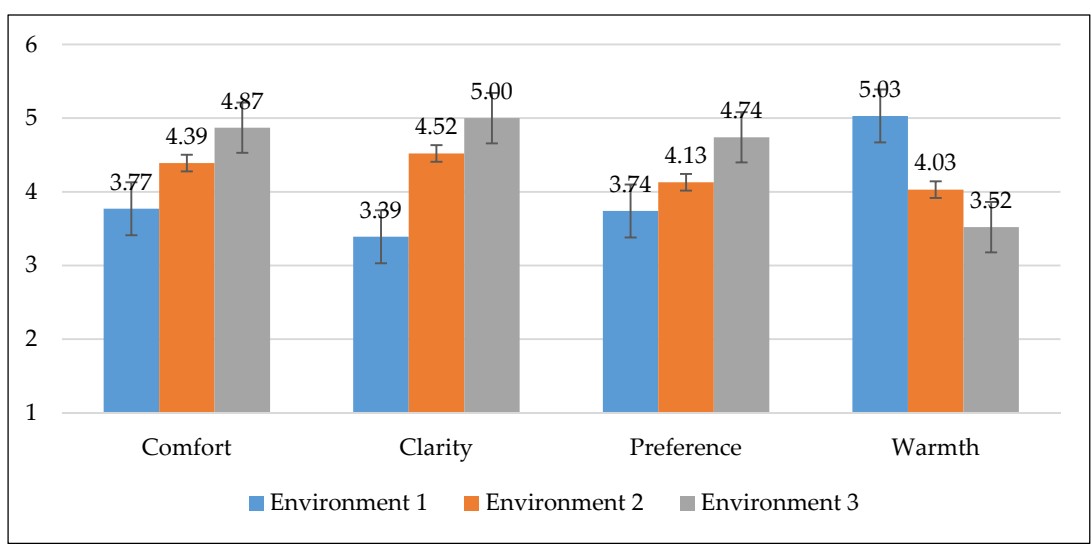

**Figure 13.** Mean value of four psychological perceptions in different environments (Environment 1 is the national museum of western art. Environment 2 is the Aichi Prefectural museum of art. Environment 3 is the Yamazaki Mazak museum of art).

It can be observed from Figure 13 that all the four psychological perception degrees are above 3, showing that the three environments are more or less suitable for people's psychological feelings. In the three environments, people hold the same views on comfort, clarity and preference. They all think that the lighting environment in the third environment is the most suitable, while the special condition of warmth is on the contrary. Environment 1 has the highest data feedback value for warmth, caused by the low CCT environment. If the value of CCTs and illuminance is low, it will result in the lack of brightness that is required to view the works and the decline in clarity. On the whole, the lighting in 3 is the most comfortable and enjoyable for the observers. The error bar for comfort, clarity and warmth in environment 1 and 2 are not at the same level as the lowest. The error bar of the four dimensions of environment 1 and environment 3 quite differ, and the psychological evaluation level of observers is quite different.

In a bid to investigate the relationship between CCT and psychological perception factors (comfort, clarity, preference and warmth), we used the correlation analysis in the mathematical statistics method and the results are shown in Table 6.

As it shows in Table 6, the significance of comfort and lighting environment was 0.047 < 0.05, and the correlation coefficient was 0.997, demonstrating that comfort was significantly correlated with the lighting environment. The significance of clarity, preference and warmth to the lighting environment was >0.1, but not much higher than 0.1, demonstrating that there was a correlation but the significance was not high. The significance of comfort and clarity, preference and warmth was >0.1, but not much higher than 0.1, indicating that there was a correlation but the significance was not high. The significance of clarity and warmth was 0.028 < 0.05, and the correlation coefficient was −0.999, demonstrating a significant negative correlation between clarity and warmth.

**Table 6.** Correlation between psychological perception and lighting environment.

| | | Correlation | | | | |
|---|---|---|---|---|---|---|
| | | Lighting-Environment | Comfort | Clarity | Preference | Warmth |
| Lighting-environment | Pearson correlation | 1 | 0.997 ** | 0.974 | 0.992 | −0.983 |
| | Significant | | 0.047 | 0.146 | 0.080 | 0.118 |
| Comfort | Pearson correlation | | 1 | 0.988 | 0.980 | −0.994 |
| | Significant | | | 0.099 | 0.127 | 0.071 |
| Clarity | Pearson correlation | | | 1 | 0.938 | −0.999 * |
| | Significant | | | | 0.226 | 0.028 |
| Preference | Pearson correlation | | | | 1 | −0.952 |
| | Significant | | | | | 0.198 |
| Warmth | Pearson correlation | | | | | 1 |
| | Significant | | | | | |

* Significant correlation at the 0.05 level (bilateral). ** Most significant correlation at the 0.01 level (bilateral).

Luo et al. studied that the most critical factor affecting the grading was illuminance, and the VM and WM [22,23] models were established. In this paper, from the perspective of a large lighting environment, we conclude that under the combination of high illuminance and high CCT, the psychological evaluation value of observers was higher.

The results show that the comfort levels found in this study are partially consistent with kruithof's rule. The lighting environment in this paper takes 3 typical illuminances and CCTs levels. That is to say, low CCT and low illuminance, middle CCT and middle illuminance, and high CCT and high illuminance. Through a series of experimental data analysis, we find that high illuminance and high CCT are the most popular lighting environment standards among observers. The lighting environment is restricted by lots of factors, and only the matching mode of CCT and illuminance is studied as the benchmark. The results are analyzed, and the four dimensions of the lighting environment are affected by illuminance and CCT.

## 4. Conclusions

In this paper, the influence of psychological perception factors on observers in three different environments was investigated via psychophysical studies. The subjective evaluation data of four kinds of psychological perceptions (comfort, clarity, preference and warmth) in different environments were investigated, and univariate analysis of variance and Pearson correlation analysis were carried out on the data. The results demonstrate that under the three lighting environments, comfort was significantly correlated to the lighting environment, the scores of comfort, clarity and preference were consistent, while the scores of clarity and warmth were negatively correlated. The best lighting environment reported by observers is high CCT and high illuminance. Also, it is the most suitable for observers to view the art museum in the best lighting environment choice.

To make the study more rigorous, and also we explain a few things. In the future, these new experiments will be performed about the next steps. The first, considering the parameters that daylighting effects artificial lighting in the art museum. The lighting environment of the three museums used in this paper is the artificial lighting environment, no used daylighting environment. In this experiment, the illuminance of three art museums' lighting environment conforms to the specification of museums, meets the requirements of annual exposure, UV, and Glare.

The second, the different lighting environments have different visual effects on young observers and older observers. In this study, the same 31 observers were used to evaluate three different art museums, so as to avoid the deviation caused by the age difference of observers. The observers of different ages will be added in future research. This result of the paper is explaining the small sample data used in this experiment to determine the response of an age group. And future research will expand the age range of the subjects, to analyze and discuss the emotional responses of different age groups in the art museum lighting environment.

The third, the blue paint on the walls in the Yamazaki Mazak Museum has any effect on the observer's perception in the space. This paper mainly discusses the impact of "lighting" on observers, "colour" has not been described and analyzed in more detail. Due to the background walls of different colours, whether the observation and response of the subjects have any effect, it needs further discussion. In the future research on the lighting environment of the art museum exhibition hall, we will give more in-depth attention and analysis in the research according to different background colours.

**Author Contributions:** Z.W. proposed the methodology as well as did the theory architecture. Y.N. did the experiment design and research direction. J.L. (Jiahui Liu) did the SPSS analysis, validation as well as the draft writing and editing. N.Z. did the theory system analysis and validation. J.L. (Jing Liang) did the validation and data curation. All authors have read and agree to the published version of the paper.

**Funding:** This research was funded by the Education Department of Liaoning Province, grant number J2019025, and funded by Dalian academy of social sciences, grant number 2019dlsky058.

**Acknowledgments:** The authors thank Liu Qiang, Michelle Zhu, the editors and reviewers to improve the paper, as well as the volunteers of taking part in our empirical experiments.

**Conflicts of Interest:** The authors declare no conflict of interest.

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
