# Peer review of "Artificial Lighting Environment Evaluation of the Japan Museum of Art Based on the Emotional Response of Observers"

_applsci, doi:10.3390/app10031121_

Round 1

Reviewer 1 Report

The topic is very interesting and try to give an answer related to lighting environment in the museums.

The paper is related to electric lighting and the title is misleading as it can be interpreted as electric/natural lighting.

The authors should be clear about this aspect.

The artworks are created and should be viewed under daylight and old or new museums try to use as much possible daylight, avoiding UV and glare issues. The conclusion of the authors that 'best lighting environment reported by visitors is high colour temperature and high illuminance' goes hand in hand with the idea of using daylighting. The author should also stress that high illuminance could be a problem for some museums objects (textile, tempera paints etc.)

So conclusion of the paper should be enhanced and the topic of daylighting should be added.

Author Response

Response to Reviewer 1 Comments

Point 1: The paper is related to electric lighting and the title is misleading as it can be interpreted as electric/natural lighting.

Response 1: Thanks to the Reviewer's suggestion, we revised the topic according to your suggestion. The lighting environment of the three museums used in this paper is the artificial lighting environment, not daylighting environment. So this title is changed to Artificial Lighting Environment Evaluation of the Japan Museum of Art Based on the Emotional Response of Observers.

Point 2: The authors should be clear about this aspect. The artworks are created and should be viewed under daylight and old or new museums try to use as much possible daylight, avoiding UV and glare issues.

Response 2: Thanks for the comments and careful reading. We fully agree. We have modified the manuscript accordingly. Please see Section 4, in blue. Most of the oil paintings in the museum were done in daylighting conditions. But this experiment is not a laboratory simulation experiment, although the spectrum of daylighting can well restore oil paintings, the illuminance and CCTs will change with the change of interior planning, window size and sunshine time of the art museum space, there are also UV and Glare issues. Since the experiments used were in three artificial lightings of art museums, in this paper only discusses the lighting environment of artificial lighting.

Point 3: The conclusion of the authors that 'best lighting environment reported by visitors is high colour temperature and high illuminance' goes hand in hand with the idea of using daylighting.

Response 3: Very good comment. This idea has been confirmed by many researchers. We have modified. Please see Section 3, in blue. The results show that the comfort levels found in this study are in part consistent with Kruithof's rule. The conclusion that the best lighting environment reported by observers is high CCT and high illuminance.

Point 4: The author should also stress that high illuminance could be a problem for some museums objects (textile, tempera paints, etc.)

Response 4: We fully agree. We have modified the manuscript accordingly. Please see Section 2.2, in blue. Dang R et al. studied the effects of illumination on inorganic pigments used in traditional paintings, high illuminance could be a problem for some museum objects (textile, tempera paints, etc.). In this experiment, the illuminance of the three art museums' lighting environment conforms to the specification of museums, meets the requirements of annual exposure, UV and Glare.

Point 5: So conclusion of the paper should be enhanced and the topic of daylighting should be added.

Response 5: We have modified and enhanced the conclusion of the manuscript. Please see Section 4, in blue. The lighting environment of the three museums used in this paper is the artificial lighting environment, no used daylighting environment. In this experiment, the illuminance of three art museums' lighting environment conforms to the specification of museums, meets the requirements of annual exposure, UV and Glare. The lighting of art museums must be concerned with the use of daylighting, but not all art museum exhibition spaces are applied by daylighting. In Japan many art museums only use artificial lighting in their exhibition spaces.

Above are my modifications. We appreciate for Reviewer's and Editor’s warm work earnestly and hope that the correction will meet with approval. Once again, thank you very much for your comments and suggestions.

Reviewer 2 Report

The paper studies the effect of different lighting environmental on the visitor’s perception of the space. Measurements of the light quality and psychophysical experiments were performed. The article lacks crucial detailing of the experimental design and settings, as well as a clear presentation of the results.

Major revision

1)           Some references are missed

2)           The meaning of the terms in the eq 1 and 2 are not listed

3)           Section 1: the object of the paper and the differences with respect of the other researches are not clear

4)           Sometimes, the CCT is reported simply as “colour temperature”

5)           Section 2.1: are the CCT, the x and y as well as the rendering index the characteristics of the light sources?

6)           Why is the R9 reported?

7)           The measurement instruments are not declared

8)           Section 2.2 explains the measurement method and, then, it should be put before the section 2.1

9)           From figures 5 to 7, it seems that the average illuminance values are evaluated on a horizontal plane. Is right? If so, as reported in the figures 5 to 7 the light strikes mainly on the vertical wall and, probably, the horizontal average illuminance values do not seem the best parameter for characterizing the illuminance levels inside a museum

10)          In section 2.3, it is not clear how the experiments were done, how the subjects were interviewed as well as what type of visual task the subjects had to perform

Author Response

Response to Reviewer 2 Comments

Point 1: Some references are missed.

Response 1: Thanks to the Reviewer's suggestion. We have modified the manuscript accordingly. The references in the manuscript were checked and uncitied references were marked.

Point 2: The meaning of the terms in the eq 1 and 2 are not listed.

Response 2: We have modified the manuscript accordingly. Please see Section 1, in blue. CP in equation (1) is the Colour preference. Ev, eq is the equivalent illuminance. ln is a logarithmic function. The mean visually scaled CP is depicted as a function of the model equation. SP in equation (2) is the Scene preference, â–³C* is saturation. Ev, eq is the equivalent illuminance. ln is a logarithmic function. The symbol S in equation (2) represents the signal of the short-wavelength sensitive human photoreceptors obtained by weighting the relative spectral power distribution of the light source with the spectral sensitivity of the S-cones and integrating over the visible wavelength range. The quantity V is obtained by weighting the relative spectral power distribution of the light source with the V(k) function and integrating over the visible wavelength range. According to the equation, the values of CP and SP are calculated to analyze and discuss the colour preference and scene preference.

Point 3: Section 1: the object of the paper and the differences with respect of the other researches are not clear.

Response 3: We have modified the manuscript accordingly. Please see abstract, in blue. The research main differs from other research in two points. The first is that the experimental lighting environment is not the laboratory simulation environment, but the real art museum lighting environment space, giving to observer the most realistic visual experience. The second is the use of three different types of art museum lighting design. The lighting design includes direct lighting, indirect lighting, and mixed lighting. Different lighting install types represent different lighting design techniques, which bring different visual feelings to the observer when viewing paintings. The different lighting designs will precisely define the art museum spatial type and optical parameters. The illuminance and CCTs of each space are set as variables, the CRI and illuminance uniformity are used as reference evaluation indicators. The main reasons for the difference in visual perception of different lighting designs in the art museum space of the art museum are analyzed through questionnaires dates. In this study, the same 31 observers response were used to evaluate three different art museums, so as to avoid the deviation caused by the age difference of observers.

Point 4: Sometimes, the CCT is reported simply as “colour temperature”.

Response 4: There is a writing error in the manuscript. We have modified the manuscript accordingly. It should be the Correlated colour temperature(CCT), and it should not be the colour temperature. The colour temperature has been changed to CCT in this manuscript.

Point 5: Section 2.1: are the CCT, the x and y as well as the rendering index the characteristics of the light sources?

Response 5: We have modified the manuscript accordingly. Please see Section 2.2, in blue. CCTs, x, and y are used to describe the CIE1931 chromaticity. The three spaces use different light sources and spectra, this method is used to define and quantify the parameters of the art museum lighting environment. And the CCTs in art museum lighting is different, which brings different visual experience and feeling to the observer. The lower the CCT, the warmer the hue of the art museum space; the higher the CCT, the cooler the hue of the art museum space, and different CCTs give different subjective feelings. In order to accurately represent the colour of the lighting environment. The study uses the x and y values to give the position of the Correlated colour temperatures(CCTs) in CIE1931 accurately and provide more detailed optical data.

Point 6: Why is the R9 reported?

Response 6: We added an explanation to the manuscript. Please see Section 2.2, in blue. Because the R9 in the art museum is a very important evaluation index. R9 is saturated red, it is an indicator of the ability of the light source to restore the redness of the object. The larger the value of R9, the higher the ability of the light source to reduce the redness of the object. The LED light source used in most museums is blue light to excite yellow phosphors to emit light. The value of the red light spectrum in the spectrum of this light source is relatively low, but the red spectrum is important in the lighting of art museums, the R9 value was used as an important indicator of evaluating the quality of the lighting environment.

Point 7: The measurement instruments are not declared.

Response 7: Very good suggestion. We have modified. Please see Section 2.1, in blue. The equipment used in the experiment to measure illuminance and CCTs is Konica Minolta CL-200A, and the CRI is measured by an Asensetek ALP-01 Pro.

Point 8: Section 2.2 explains the measurement method and, then, it should be put before the section 2.1.

Response 8: Thanks for the comments and careful reading. We fully agree. We have modified the manuscript. Put the description of measurement method in section 2.1, and label its references.

Point 9: From figures 5 to 7, it seems that the average illuminance values are evaluated on a horizontal plane. Is right? If so, as reported in the figures 5 to 7 the light strikes mainly on the vertical wall and, probably, the horizontal average illuminance values do not seem the best parameter for characterizing the illuminance levels inside a museum.

Response 9: We fully appreciate the helpful comments. We have modified the manuscript. Please see Section 2.2, in blue. In this section, the representative paintings of the three art museum exhibition and the background wall illumination data were added. Since the exhibition hall space uses three different lighting designs, the illumination and uniformity of the background walls are also different, such as Figure 6, Figure 8 and Figure 10. Added the content of the classification of the architectural space layout of the typical exhibition hall, indicating that observer in the exhibition viewing space, mainly in the rest space and transportation space, use the ground illumination to evaluate the lighting environment. And in the viewing area, the representative painting works and the background wall illumination are used to evaluate the lighting environment.

Point 10: In section 2.3, it is not clear how the experiments were done, how the subjects were interviewed as well as what type of visual task the subjects had to perform.

Response 10: Very good suggestion. We have modified. Please see Section 2.1, in blue. A total of 31 observers response were surveyed in this experiment to visit three art museums, including 16 males and 15 females. Table1, Figure 11 and Figure 12 was added to describe the experiment.

The experimental process is as below:

Introduce the content and process of the art museum lighting experiment to the observer. Visual testing. Test the observer for visual defects or color blindness. Environmental adaptation. The observer first adapts to the museum lighting environment under a typical space for about 2 minutes. The experiment should allow the observers to whole experience the art museum lighting environment of the exhibition space, and evaluate and feel the lighting environment of the viewing space, traffic space and rest space through observers' vision. Main experiment. Through the moving, viewing and staying of the observer in the exhibition space visual task is completed, the process takes about 5 minutes. Next, observers viewing the typical painting lighting environment in the viewing space, observe the oil paintings and feel the lighting environment in the viewing space, the process takes about 3 minutes.

Figure 12 shows the observation distance of the observers when viewing the oil painting. A limit is set before each painting to control the viewing distance to ensure that the observer's sightline is at the best viewing angle and distance. The typical size of the oil painting used in this experiment is suitable, so the observers are looking to have the optimal visual distance.

Subjective evaluation. The observer makes a subjective evaluation of the current lighting environment in the art museum, the observer fills in the corresponding score according to the questionnaire. The experimental process obtained subjective response data through four evaluation dimensions (comfort, clarity, preference, and warmth) from observers. And analyzes the influence of the actual spatial lighting parameters of art museum buildings on observers' psychological emotions. Museum lighting environmental changes. Repeat steps 3~5 until the experimental data are collected.

Above are my modifications. We appreciate for Reviewer's and Editor’s warm work earnestly and hope that the correction will meet with approval. Once again, thank you very much for your comments and suggestions.

Reviewer 3 Report

Lighting plays an important role with regard to museum design and any visitor perception and response. Thus, this article is an important contribution to  a basic aspect of exhibition and museum conception and arrangement. The authors chose three Japanese museums with different lighting environments and selected 31 observers, aged 20-24 years. The psychological perception factors of comfort, clarity, preference and warmth are well chosen, the results interesting and helpful for any museum design activities.

What might be critisized is the selection of observers that does not offer any comparative approach. It has already become clear that, in particular, young and elderly observers react differently to any kind of museum design, also with regard to the lighting environment. Therefore, it would have been advisable to select different groups of observers and compare their perception and emotional response. If this cannot be done now, it should, at least, be mentioned that further research and analysis will also have to concentrate on such comparisons of the reaction of distinct observers.

Author Response

Response to Reviewer 3 Comments

Point 1: What might be critisized is the selection of observers that does not offer any comparative approach. It has already become clear that, in particular, young and elderly observers react differently to any kind of museum design, also with regard to the lighting environment. Therefore, it would have been advisable to select different groups of observers and compare their perception and emotional response. If this cannot be done now, it should, at least, be mentioned that further research and analysis will also have to concentrate on such comparisons of the reaction of distinct observers.

Response 1: Thanks to the Reviewer's suggestion, we have modified the manuscript accordingly. Please see Section 4, in blue. Thanks for the comments and careful reading. We fully agree. It is true that different lighting environments have different visual effects on young observers and older observers. The observers of different ages will be added in future research. In this study, the same 31 observers were used to evaluate three different art museums, so as to avoid the deviation caused by the age difference of observers. This result of the paper is explaining the small sample data used in this experiment to determine the response of an age group. And future research will expand the age range of the subjects, to analyze and discuss the emotional responses of different age groups in the art museum lighting environment.

Above are my modifications. We appreciate for Reviewer's and Editor’s warm work earnestly and hope that the correction will meet with approval. Once again, thank you very much for your comments and suggestions.

Reviewer 4 Report

Well presented and will be of interest to a number of readers.

I have only minor suggestions for clarity improvement:

Move the paragraph directly above "section 2. Experiment" (starts with the line "This experiment differs from the control variable...". Move this paragraph up earlier in the paper, and then follow with the review of the previous studies that only consider one dimension, while your study considers 4 dimensions. 

The paragraph also needs to be edited for clarity. This key sentence of your thesis is confusing as written: "Compared with other studies on the influence of a certain dimension on visual feedback via the control variable method, this paper focuses on the comprehensive analysis on account of examples." ??

Lots of good charts and graphs, very helpful. Only suggestion: on Figure 8 need to clarify which is environment 1 vs. 2 vs. 3. You have switched the order of the museums in the course of the paper, so not exactly sure which is which by figure 8. 

one unrelated question: does the blue paint on the walls in the Yamazaki Mazak Museum have any effect on people's perception in the space? Many museums have moved away from strictly white walls, and it might be useful to know whether that has an impact on perception....

Author Response

Response to Reviewer 4 Comments

Point 1: Move the paragraph directly above "section 2. Experiment" (starts with the line "This experiment differs from the control variable...". Move this paragraph up earlier in the paper, and then follow with the review of the previous studies that only consider one dimension, while your study considers 4 dimensions. 

Response 1: Thanks to the Reviewer's suggestion, we have modified the manuscript accordingly. Please see Section 2, in blue. The main survey of the comfort level is the coordination of the overall space light and shadow of the museum exhibition space, as well as the degree of psychological colour. Clarity checks whether the details and texture of the exhibits can be clearly displayed, which is satisfactory. Preference is that the overall artistic effect of lighting is outstanding, there are wonderful lighting performance effects, and whether the lighting environment in which you are feeling the like. The warmth refers to the contrast of warmth and coldness on the senses, whiter means colder and yellower means warmer.

Point 2: The paragraph also needs to be edited for clarity. This key sentence of your thesis is confusing as written: "Compared with other studies on the influence of a certain dimension on visual feedback via the control variable method, this paper focuses on the comprehensive analysis on account of examples." ?

Response 2: Thanks a lot for the careful reading. We have modified the manuscript. Please see Section 2, in blue. The paper is mainly to study the four different dimensions in a large space environment for observers to experience the psychological feelings of museum lighting, it is not a separate investigation of CCT and illuminance changes, but specific indicators in the actual museum environment are used for investigation and research. Correlation analysis and one-way analysis of variance were used to analyze the differences in four dimensions under three different lighting environments.

Point 3: Lots of good charts and graphs, very helpful. Only suggestion: on Figure 8 need to clarify which is environment 1 vs. 2 vs. 3. You have switched the order of the museums in the course of the paper, so not exactly sure which is which by figure 8. 

Response 3: Very good suggestion. We have modified. Please see Section 3, in blue. Figure 8 relabels the three lighting environments and checks the description in the text; environment 1 is the National museum of western art, environment 2 is the Aichi Prefectural museum of art, and environment 3 is the Yamazaki Mazak museum of art.

Point 4: one unrelated question: does the blue paint on the walls in the Yamazaki Mazak Museum have any effect on people's perception in the space? Many museums have moved away from strictly white walls, and it might be useful to know whether that has an impact on perception....

Response 4: Thanks for the comments and careful reading. We fully agree. We have modified the manuscript accordingly. Please see Section conclusions, in blue. A definition of the background "colour" will be added in future research to describe and define the lighting environment of the space more accurately. In future research on the lighting environment of the art museum exhibition hall, we will give more in-depth attention and analysis in the research according to different background colours.

Above are my modifications. We appreciate for Reviewer's and Editor’s warm work earnestly and hope that the correction will meet with approval. Once again, thank you very much for your comments and suggestions.

Round 2

Reviewer 2 Report

The manuscript has been significantly improved and now is ready for publication in Applied Sciences.